# Phase-Specific Changes in Rate of Force Development and Muscle Morphology Throughout a Block Periodized Training Cycle in Weightlifters

**DOI:** 10.3390/sports7060129

**Published:** 2019-05-28

**Authors:** Dylan G. Suarez, Satoshi Mizuguchi, William Guy Hornsby, Aaron J. Cunanan, Donald J. Marsh, Michael H. Stone

**Affiliations:** 1Center of Excellence for Sport Science and Coach Education, Department of Sport, Exercise, Recreation, and Kinesiology, East Tennessee State University, Johnson, TN 37614, USA; harahara10@hotmail.com (S.M.); aaron.cunanan@gmail.com (A.J.C.); dmarsh790@gmail.com (D.J.M.); 2Department of Coaching and Teaching Studies, West Virginia University, Morgantown, WV 26505, USA; william.hornsby@mail.wvu.edu (W.G.H.); stonem@etsu.edu (M.H.S.)

**Keywords:** periodization, athlete monitoring, muscle, rate of force development, isometric mid-thigh pull

## Abstract

The purpose of this study was to investigate the kinetic and morphological adaptations that occur during distinct phases of a block periodized training cycle in weightlifters. Athlete monitoring data from nine experienced collegiate weightlifters was used. Isometric mid-thigh pull (IMTP) and ultrasonography (US) results were compared to examine the effects of three specific phases of a training cycle leading up to a competition. During the high volume strength-endurance phase (SE) small depressions in rate of force development (RFD) but statistically significant (*p* ≤ 0.05) increases in vastus lateralis cross-sectional area (CSA), and body mass (BM) were observed. The lower volume higher intensity strength-power phase (SP) caused RFD to rebound above pre-training cycle values despite statistically significant reductions in CSA. Small to moderate increases only in the earlier RFD time bands (<150 ms) occurred during the peak/taper phase (PT) while CSA and BM were maintained. Changes in IMTP RFD and CSA from US reflected the expected adaptations of block periodized training phases. Changes in early (<100 ms) and late (≥150 ms) RFD time bands may not occur proportionally throughout different training phases. Small increases in RFD and CSA can be expected in well-trained weightlifters throughout a single block periodized training cycle.

## 1. Introduction

Competitive success in the sport of weightlifting relies on the kinetic and kinematic abilities of the athlete. However, after a few months to years of training weightlifting technique tends to become highly stable [1,2], while the weight lifted and power outputs continue to increase [3]. There is also ample evidence that suggests weightlifting success is heavily dependent on the magnitude and rate of force development (RFD) generated by the lifter [4,5,6]. Therefore, the performance of more advanced weightlifters is likely primarily determined by the capacity to generate high forces, RFD, and peak power outputs [7,8] during the competitive lifts. These characteristics are often specifically targeted through unique training periods that aim to create certain adaptations to the neuromuscular system. The ability to assess both the magnitudes and timelines of which these adaptations occur can be beneficial for designing the training of strength and power athletes.

Weightlifters benefit from only participating in a few major competitions per year allowing for certain training phases to be dedicated to the development of specific adaptations (e.g., hypertrophy, maximum strength, speed, etc.). Block periodization can serve as a framework for sequentially eliciting these adaptations across training phases, culminating in a peak where the athlete has the highest potential of success on the day of competition [9]. This strategy is conducted in phases often referred to in the literature as accumulation, transmutation, and realization [10]. This sequence of training phases is intended to initially emphasize the development of work capacity and force generating potential in order to potentiate the following phases of more specific training. DeWeese et al. [11] suggests that the training process for a strength-power athlete not only requires an appropriate stimulus for adaptation but also benefits from an appropriate method of assessing progress (i.e., monitoring).

The isometric mid-thigh pull (IMTP) is a commonly used method to both assess the kinetic ability of an athlete as well as monitor changes in their performance potential throughout a training period [12]. The IMTP is especially valuable for the monitoring of weightlifters since it provides the opportunity to safely measure important performance variables, such as peak force (PF) and RFD in a sport-specific position. Strong correlations (r ≥ 0.70) have been observed between these variables and weightlifting performance [12,13,14]. However, research suggests that RFD is more closely related to most athletic tasks [15,16] and is more sensitive to fatigue [12,17]. Haff et al. [18] reported that calculating RFD using specific time bands results in higher reliability than quantifying peak RFD values. Additionally, these various RFD time bands have been suggested to be governed by different physiological mechanisms and therefore may respond distinctively to various training phases. For example, earlier RFD time bands (<100 ms from onset) have been suggested to be influenced to a greater degree by neural factors and intrinsic muscle properties [19,20,21,22,23]. Conversely, later RFD time bands (>100 ms from onset) are more closely related to maximal muscle strength and size [24,25,26]. Since block periodized training consists of distinct phases that emphasize certain physical qualities, the RFD time bands may be affected differently. For instance, a concentrated load of strength-endurance over multiple weeks often results in depressions in measures of power and speed in trained athletes [27], but once the athlete returns to regular training increases potentially above previous values (i.e., supercompensation) can occur [12,28,29,30]. Furthermore, realization phases apply a substantial decrease in training volume with a corresponding increase or maintenance in training intensity aimed at substantially decreasing neuromuscular fatigue and inducing certain adaptations such as shifts to faster fiber types [31,32,33]. Adaptations commonly associated with these phases may potentially be most apparent in the earlier RFD time bands, but need to be further investigated.

Both PF and RFD are influenced by the size, architecture, and composition of muscle fibers [23,26,34,35,36]. Ultrasonography (US) provides a non-invasive method for assessing and monitoring muscle qualities like muscle thickness (MT), cross-sectional area (CSA), pennation angle (PA), and fascicle length (FL) [37,38,39]. Reported changes in these variables throughout training periods are mixed and seem to be dependent on the style of training [36,40,41,42]. Increases in the size of a muscle from resistance training has been well established. However, the extent to which a single three to four weeks hypertrophy phase as is often seen in block periodized programs, results in increased muscle size in well-trained athletes is unclear. Also, less well understood are the timelines of which changes to muscle morphology occur throughout different phases of the training cycle. Additionally, muscle hypertrophy is highly dependent on training volume [43,44], and studies investigating changes in muscle size during periods of reduced training volume have observed concomitant reductions in body mass and muscle size [45,46]. Therefore, weight class athletes who often deliberately lose body mass leading up to a competition may be at a greater risk of muscle loss during realization phases. 

Coaches and sport scientists of any strength-power sport can benefit from further clarification into the expected magnitudes and timelines of adaptation to block periodized training. Therefore, this investigation sought to better understand the kinetic and morphological adaptations that occur during distinct phases of a training cycle in advanced strength athletes using the IMTP and US as longitudinal athlete monitoring tools. 

## 2. Materials and Methods

This study was a comparison of pre- and post-block testing results from three specific training phases throughout a single macrocycle leading up to a competition. The initial training phase (T1–T2) consisted of three weeks of high volumes and low to moderate relative intensities, termed a Strength-Endurance Phase (SE). The second phase of training (T2–T3) consisted of four weeks of moderate volumes at higher intensities, termed a Strength-Power Phase (SP). The final block of training (T4–T5) occurred at the very end of each macrocycle where the athletes underwent a single week of a sharp increase in volume (Overreach), followed by a three-week taper of low volume and moderate intensities, termed a Peak/Taper Phase (PT).

Because of variations in the subjects training age and performance levels, the length of the athlete’s macrocycles varied (~4–7 months) depending on the time between their most important competitions. Therefore, for the purposes of this study pre- and post-block testing results from three distinct training phases were selected for each athlete (Figure 1). Each training phase closely resembled the relative volumes and intensities of the other athletes and took place as the very first and second blocks of the macrocycle and the very last. Ultrasound testing sessions were conducted at the end of the final training week at least 24–48 h after the previous training session. Testing conducted with the IMTP occurred on Monday mornings approximately 48 h after the last training session (Saturday) and before beginning a new block of training, or on Wednesday morning after the peak/taper block (T5) to allow dissipation of fatigue from travel to and back from competition the previous weekend. All testing sessions occurred after a planned week of reduced training volume.

### 2.1. Athletes

Athlete monitoring data from a total of nine experienced collegiate weightlifters was used for analysis (Table 1). All nine of these athletes had competed at least at the university national level, three at the senior national level, and one had previously competed internationally as a junior and university world team member. All athletes were familiar with the testing procedures, and the data were collected as part of an ongoing athlete monitoring program. The study was approved by the East Tennessee State University Institutional Review Board (#c0218.18sw) and the athletes provided consent for their monitoring data to be used.

### 2.2. Training

Training was organized in a four-day per week push-pull layout, and an example training plan is summarized in Table 2 and Table 3. The training program was designed, implemented, and adjusted by nationally certified coaches, and the researchers had no influence on the training itself.

### 2.3. Hydration

Before IMTP and ultrasound testing sessions, the hydration levels of the athletes were estimated using a handheld refractometer (Atago 4410 PAL-10S, Tokyo, Japan) to calculate urine specific gravity (USG) on a scale ranging from 1.000 to 1.060. If the athletes USG registered as ≥1.020, they had to continue to rehydrate until they registered below 1.020. This was performed to control for dehydration having any adverse effects on the athletes’ performance [48] and the overall testing results.

### 2.4. Warm-Up

Isometric mid-thigh pull testing was preceded by a standardized warm-up protocol consisting of 25 jumping jacks followed by a set of five dynamic mid-thigh pulls with a 20-kg bar. Athletes then performed three sets of five repetitions, with approximately one-min rest between sets, of dynamic mid-thigh pulls with 60 kg (males) or 40 kg (females).

### 2.5. Isometric Mid-Thigh Pull

Isometric mid-thigh pull testing was performed standing on dual force plates (Rice Lake Weighing Systems, Rice Lake, WI, USA; 1000 Hz sampling rate) inside of a custom-designed power rack that allows adjustment to the desired bar height. Athletes began the testing by assuming a mid-thigh pull position for which they were already familiar performing both in training and for testing (Figure 2). Knee angle was measured to be 125 ± 5 degrees (measured using a handheld goniometer), and the lifter was then instructed to perform a 50% effort warm-up isometric pull. After a brief rest, the athlete performed another warm up pull at 75% and was then secured to the bar with both lifting straps and athletic tape. Athletes were instructed to “pull as fast and hard as possible” beforehand. For the trials, verbal instruction was given to get into position and apply a steady amount of pre-tension to the bar to reduce slack in the body, and to help minimize a countermovement. Once a consistent force trace was observed by the tester a verbal countdown of “3, 2, 1 pull” was given with loud verbal encouragement given until the tester noticed a plateau or decrease in force. Athletes then received 90–120 s of seated rest before reattempting. Additional trials were performed if there was a >250 N difference in peak force from the first attempt. The force trace was analyzed by the same investigator using custom designed lab view software (National Instruments, Austin, TX, USA). The mean of the best two attempts for PF as well as RFD time intervals of 0–50 ms (RFD50), 0–100 ms (RFD100), 0–150 ms (RFD150), 0–200 ms (RFD200), and 0–250 ms (RFD250) was used.

### 2.6. Ultrasonography

A 7.5 MHz ultrasound probe (LOGIQ P6, General Electric Healthcare, Wauwatosa, WI, USA) was used to measure CSA, MT, FL, and PA of the vastus lateralis (VL). Measurements were taken in a standing position as described by Wagle et al. [49], as this position has been shown to correlate better with both isometric and dynamic performance. The tester identified and marked 50% of the distance between the greater trochanter and the lateral epicondyle of the right leg. Three MT images were then taken five centimeters anteromedial to the mid-femur mark. The best image from the three was selected for analysis, and the mean of three MT and PA measurements was taken from the first, second, and third portions of the image. Three CSA images were attained by using a panoramic image sweep perpendicular to the VL muscle at the mid-femur mark. CSA was then determined by selecting two out of the three images that best displayed the region of interest and using an image processing software (ImageJ 1.52a, National Institutes of Health, Bethesda, MD, USA) to trace the intermuscular area (Figure 3a). Lastly, FL was estimated by calculating MT∙sin(PA)^−1^ (Figure 3b). The US technician remained the same throughout all five testing sessions, and all images were analyzed by a single researcher on the same computer.

### 2.7. Statistical Analyses

All data has been represented as mean ± SD. A Shapiro–Wilk normality test was used to verify if the data were normally distributed. One-way and two-way repeated measures analysis of variance (ANOVA) were performed to determine the effects of training phase (one-way) and the main and interaction effects of phase and RFD time bands (two-way) on the measured variables. Statistical effects were followed up with post-hoc pairwise comparisons with Bonferroni adjustment. Effect sizes (Cohen’s d) and 95% confidence intervals were calculated to better provide population parameter estimates of mean change and to infer practically meaningful changes. These changes were interpreted using the following scale: 0.0–0.2 (trivial); 0.2–0.6 (small); 0.6–1.2 (moderate); 1.2–2.0 (large); 2.0–4.0 (very large; 4.0+ (nearly perfect) [50]. The critical alpha of 0.05 was used for all null hypothesis testing unless familywise error was expected. Statistical analyses were performed using SPSS 25.0 (IBM Corp., Armonk, NY, USA), Microsoft Excel 2016 (Microsoft Corporation, Redmond, WA, USA), and RStudio (Version 1.1.383; RStudio, Inc., Boston, MA, USA).

## 3. Results

### 3.1. Isometric Mid-Thigh Pull

No statistical main or interaction effects (*p* ≤ 0.05) occurred for any of the IMTP variables (Table 4). During the SE phase (T1–T2) there were trivial to small decreases in RFD50 (d = −0.12, 95% CI [−1.04 to 0.81), RFD100 (d = −0.43, [−1.37 to 0.53]), RFD150 (d = −0.35, [−1.28 to 0.32]), RFD200 (d = −0.27, [−1.20 to 0.67]), and RFD250 (d = −0.22, [−1.14 to 0.72]). During the SP phase (T2–T3) there were moderate increases in RDF50 (d = 0.98, [−0.10 to 2.01), RFD100 (d = 1.05, [−0.05 to 2.09]), RFD150 (d = 0.68 [−0.33 to 1.65]), RFD 200 (d = 0.60, [−0.39 to 1.56]), and a small increase in RFD250 (d = 0.52, [−0.46 to 1.46]). Lastly, the PT phase (T4–T5) resulted in moderate increases in RFD50 (d = 0.78, [−0.25 to 1.76]), RFD100 (d = 0.80, [−0.23 to 1.79]), and RFD150 (d = 0.60, [−0.39 to 1.56]) only. When comparing RFD after each training phase to pre-training cycle values there was a moderate increase in RFD50 (d = 0.91. [−0.15 to 1.93]) and RFD100 (d = 1.09, [−0.01 to 2.15]), and small increases in RFD150 (d = 0.58, [−0.41 to 1.53]), RFD200 (d = 0.40, [−0.56 to 1.34]) and RFD250 (d = 0.28, [−0.67 to 1.20]) from T1–T3. There were also moderate increases in RFD50 (d = 0.87, [−0.18 to 1.87]) and RFD100 (d = 0.69, [−0.32 to 1.66]), and a small increase in RFD150 (d = 0.40, [−0.56 to 1.33] from T1–T5. Changes in PF throughout every timepoint were trivial (d = −0.23 to 0.03). Within session intraclass correlation coefficient (ICC) and coefficient of variation (CV) for each variable were: PF (ICC = 0.99, CV = 2%), RFD50 (ICC = 0.86, CV = 15%), RFD100 (ICC = 0.85, CV = 13%), RFD150 (ICC =0.91, CV = 10%), RFD200 (ICC = 0.93, CV = 8%), RFD250 (ICC = 0.94, CV = 7%).

### 3.2. Ultrasonography

The ANOVA revealed a statistically significant effect of time on CSA (*p* ≤ 0.001) and BM (*p* = 0.01). During the SE phase (T1–T2) a statistically significant increase in CSA (*p* = 0.004; d = 1.90, [0.53 to 3.21]) and BM (*p* = 0.007; d = 1.6, [0.38 to 2.90]) occurred. During the SP phase (T2–T3) CSA significantly decreased (*p* = 0.009; d = −1.61, [−2.82 to −0.34]) while BM remained mostly unchanged (*p* = 0.08; d = −0.37, [−1.3 to 0.57]). Both CSA (*p* = 0.03; d = 1.19, [0.06 to 2.27]) and BM (*p* = 0.02; d = 2.10, [0.65 to 3.50]) at T3 remained significantly higher than T1. No statistically significant change in CSA (*p* = 0.83; d = −0.10, [−1.02 to 0.83]) or BM (*p* = 0.96; d = −0.02, [−0.94 to 0.89]) occurred during the PT phase (T4–T5). Overall from T1–T5 there was a non-statistically significant but moderate increase in CSA (*p* = 0.19; d = 0.67, [−0.34 to 1.63] and BM (*p* = 0.79; d = 0.94, [−0.12 to 1.96]. There was a moderate increase in MT (d = 1.03, [−0.06 to 2.08]) during the SE phase, followed by a moderate decrease after the SP phase (d = −0.81, [−1.80 to 0.23]), and a trivial decrease during the PT phase (d = −0.14, [−1.06 to 0.79]). From T1–T5 the overall increase in MT was small (d = 0.34, [−0.61 to 1.27]). No statistically significant change in PA or FL was observed however a moderate increase in FL (d = 0.70, [−0.30 to 1.68]) and a corresponding small decrease in PA (d = −0.58, [−1.53 to 0.41]) occurred between T1–T5. Within session intraclass correlation coefficient (ICC) and coefficient of variation (CV) for each variable were: CSA (ICC = 0.99, CV = 1%), MT (ICC = 0.96, CV = 2%), PA (ICC = 0.83, CV = 9%), FL (ICC = 0.73, CV = 9%).

## 4. Discussion

The primary finding of this investigation was that changes in IMTP RFD and CSA from US reflect the expected adaptations to block periodized training phases. The SE phase resulted in slight depressions in RFD (Figure 4a), likely due to high levels of accumulated fatigue, but also caused significant increases in CSA (Figure 5a). During the SP phase, all RFD time bands rebounded above previous values (Figure 4b), and CSA decreased, but remained higher than baseline. After the PT phase only the earlier (≤150 ms) RFD time bands increased (Figure 4c) and CSA was maintained. 

In most cases where changes were observed the calculated confidence intervals suggested the responses could range from very large improvements to small decrements in performance. The only clear group changes occurred in CSA from T1–T2, T2–T3, and T1–T3. Meaning it is very likely a three to four weeks SE phase first results in small to large increases in CSA, followed by a reduction during the following phase, but a maintenance above original values (Figure 5a). This is possibly explained by Damas et al. [51] observations of early increases in CSA being primarily attributed to muscle swelling. Damage to the muscle from high volume training during the SE phase would also explain the trend of decrements in force production that were observed, and that has been reported previously [12]. After the SP phase, the RFD values in all time bands rebounded to above pre-training cycle values (Figure 4b). The significant increase in CSA and BM, the likely reduction in muscle damage from the lowered volume, and the reintroduction of higher intensities all likely contributed to this supercompensation effect. Although, not statistically significant the values of CSA, MT, and BM progressively decreased between T2–T5 (Figure 5), indicating that the increases in muscle size that occurred early in the training phase gradually decreased across the rest of the training cycle as the athlete’s body mass lowered leading up to the competition. No statistically significant change in CSA or MT occurred during the PT phase most likely because this group did not significantly alter their body mass within this short period. Seven out of the nine lifters experienced increases in CSA after the training cycle while only four ended with a greater body mass (Figure 5c). Therefore, increases in muscle size are more likely to occur in athletes who have room within their weight class to gain body mass throughout a training cycle, but may still be possible in those that maintain their weight and improve their body composition. There were no clear effects of any individual training phase on muscle architecture however there was a moderate increase in FL and a small decrease in PA from T1–T5 (Table 4). Similar changes in FL throughout a periodized training period have been observed in athletes [36,41,42] and may be representative of a shift to higher velocity movements across the training cycle.

As has been observed previously, PF remained very stable throughout the entire training cycle and RFD exhibited a much greater plasticity [12]. Changes in RFD did not at any point reach statistical significance but trends for the different training phases were observed in most of the time bands. Previous research has suggested that early RFD time bands are more closely related to neural function and late RFD is more commonly associated with maximal muscle strength [25]. Larger effects throughout each training phase in this study occurred in RFD50, RFD100, and RFD150. The lack of more substantial effects in the later RFD time bands is not too surprising as maximal force abilities, measured by PF, did not change considerably at any point. 

A major limitation of this study was the post-PT testing session occurred several days after the theoretical “peak” would have occurred. It is a common observation within our laboratory that fatigue from the competition, travel, and possible emotional let-down after the meet negatively influences these testing sessions. Additionally, due to differences in the length of the athlete’s macrocycles, it is difficult to determine the effects of what occurred between the SP and PT phase (T3–T4) had on the final two testing sessions. Therefore, it is challenging to properly compare the results at T5 to the other time points. Increases in the earlier RFD time bands (≤150 ms) were still observed between T4–T5 so it is possible that on the day of competition RFD may have been at its highest point in all time bands. But further research must be conducted to better elucidate the effects of PT phases on early versus late RFD.

Research into the adaptations that occur in well-trained strength athletes who compete in individual sports is often difficult because the timelines of the training programs may differ dependent on the competitions they have qualified for. Therefore, within the literature many insights into training adaptations in individual sport athletes are conducted as case studies. A novel aspect of this study was the grouping of athletes pre and post monitoring results together based off of similar training phases. This allowed for observations to be made on a larger sample size of well-trained subjects making the results more applicable to a wider range of athletes. Coaches and sport scientists may benefit from the use of a similar methodology in order to better evaluate the effectiveness of a training program on a group of athletes whose training cycles may not line up. 

The overall increases in muscle size and RFD throughout the entire study were not statistically significant. However, effect sizes and confidence intervals suggest small to moderate effects occurred in most variables. Additionally, all of the subjects in this study were well-trained experienced strength athletes, and the baseline values at T1 were collected after the previous training cycle, and not after a period of detraining. Therefore, it can be expected that the changes that occurred during this macrocycle would occur throughout most training cycles in athletes at this level. In the context of a long-term athlete development plan then, these effects may be quite meaningful as they could be compounded over several collective macrocycles.

## 5. Conclusions

The plasticity of RFD in addition to its greater relevance to most athletic tasks [15,16] make it a superior monitoring variable than PF. In well-trained strength athletes, PF may be more effectively used for monitoring long term changes in maximal force producing abilities while RFD provides a more comprehensive indication of the current performance potential of the athlete. Since IMTP RFD is such a valuable metric, greater attention should be placed on obtaining trials that not only display consistent PF values but also a similarity in the slope of the force-time curve. Additionally, it is important to measure RFD across multiple time bands because changes in early and late RFD may not occur proportionally. Both RFD and CSA from US seemed to reflect the expected general adaptation trend of each training phase. Therefore, coaches and sport scientists interested in assessing the kinetic and morphological adaptations to periodized training can benefit from these monitoring tools. Based on the results of this study small increases in RFD and muscle size can be expected throughout a single block periodized training cycle in well-trained weightlifters. Therefore, these results appear to support the long-term use of block periodization alongside an effective monitoring program.

## Figures and Tables

**Figure 1 sports-07-00129-f001:**
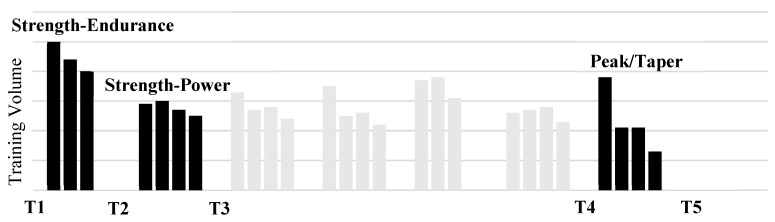
Example Macrocycle and Testing Schedule.

**Figure 2 sports-07-00129-f002:**
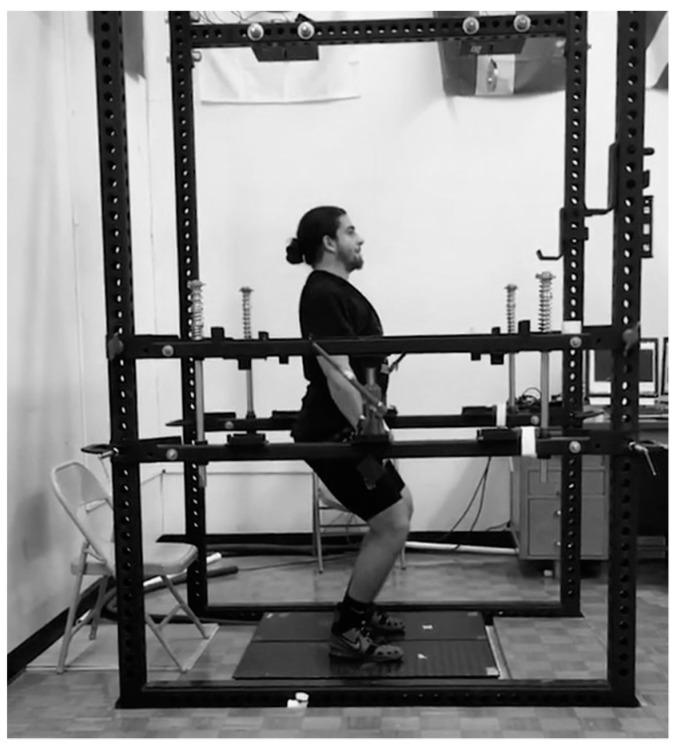
Isometric Mid-Thigh Pull Position.

**Figure 3 sports-07-00129-f003:**
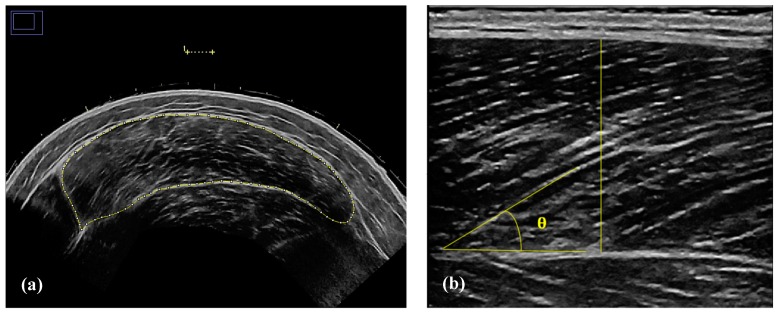
(**a**) Cross-sectional area measurement. (**b**) Muscle thickness and pennation angle measurement.

**Figure 4 sports-07-00129-f004:**
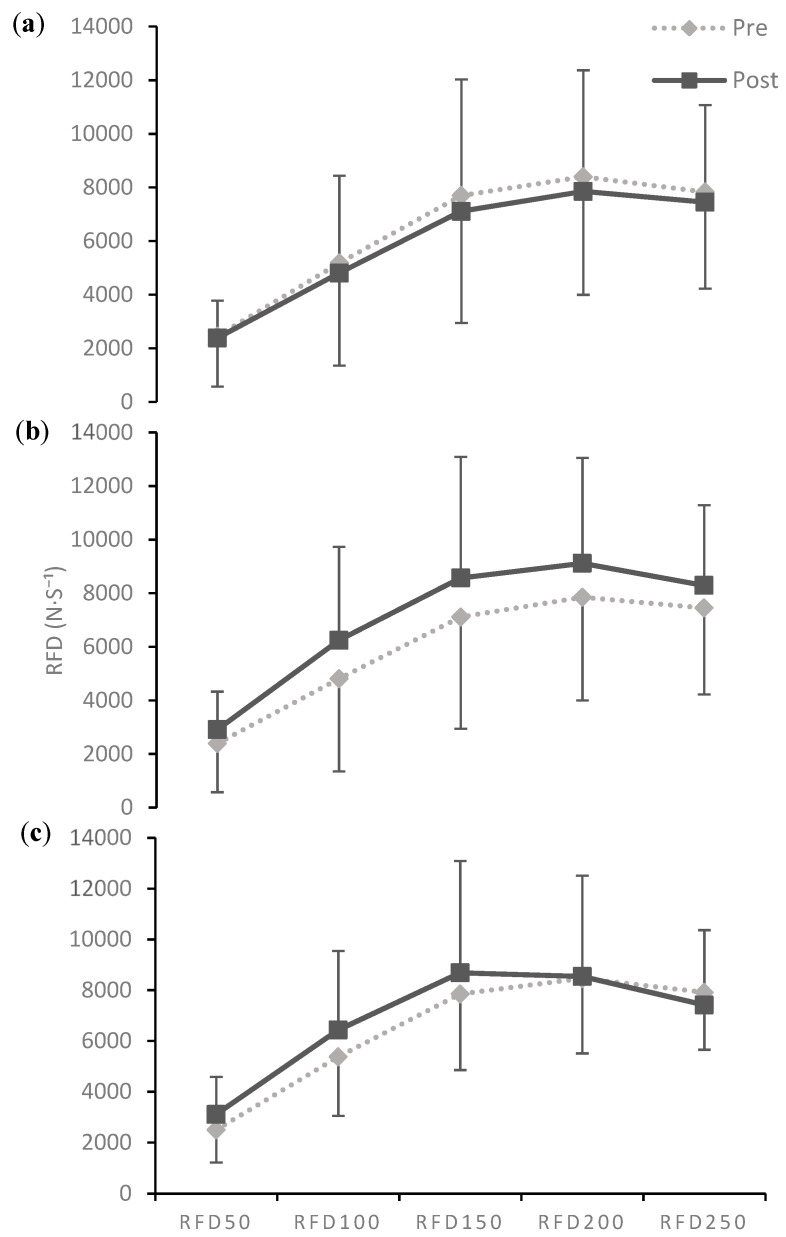
Phase-specific changes in rate of force development. (**a**) Strength-endurance phase (T1–T2); (**b**) Strength-power phase (T2–T3); (**c**) Peak/taper phase (T4–T5); mean ± SD.

**Figure 5 sports-07-00129-f005:**
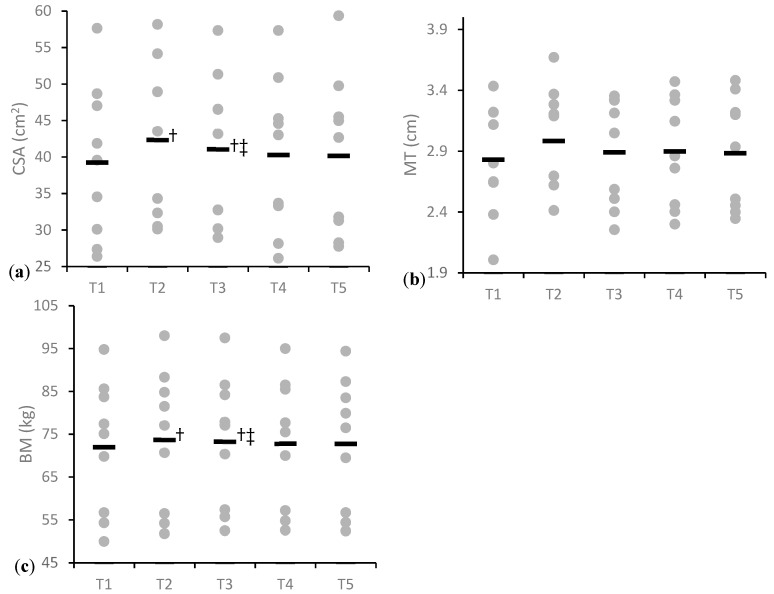
Muscle size and body mass throughout each time point. (**a**) Cross-sectional area; (**b**) Muscle thickness; (**c**) Body mass. Gray dots represent individual subjects and black line represents the group mean. † Significantly different from the previous timepoint (*p* ≤ 0.05). ‡ Significantly different from T1 (*p* ≤ 0.05).

**Table 1 sports-07-00129-t001:** Summary of Subject Characteristics (mean ± SD).

Sex	Age (years)	Height (cm)	BM (kg)	BF (%)	RT Age (years)	WL Age (years)	Snatch (kg)	C & J (kg)	IPF (N)
Males	22.4 ± 1.6	169.9 ± 3.8	83.7 ± 7.0	11.7 ± 3.0	5.4 ± 1.0	3.8 ± 0.4	117.6 ± 8.2	147.8 ± 13.6	6147.2 ± 860.6
Females	20.5 ± 2.6	157.3 ± 4.0	57.6 ± 7.2	16.8 ± 1.9	7.0 ± 3.1	6.5 ± 3.2	69.3 ± 8.0	90.8 ± 10.1	4431.0 ± 609.7

Note: Males (n = 5), Females (n = 4), BM = body mass, BF = body fat, RT = resistance training, WL = weightlifting, C & J = clean and jerk, IPF = isometric peak force.

**Table 2 sports-07-00129-t002:** Example Training Plan.

Phase	Week	Sets × Reps	Daily Intensities (M, W, Th, S)
SE	1	3 × 10	M, M, VL, VL
SE	2	3 × 10	MH, MH, L, L
SE	3	3 × 10	L, L, VL, VL
SP	1	3 × 5 (1 × 5)	M, M, L, VL
SP	2	3 × 5 (1 × 5)	MH, MH, L, VL
SP	3	3 × 3 (1 × 5)	H, H, L, VL
SP	4	3 × 2 (1 × 5)	MH, L, VL, VL
PT	1	5 × 5 (1 × 5)	MH, M, L, VL
PT	2	3 × 3 (1 × 5)	M, MH, VL, VL
PT	3	3 × 3 (1 × 5)	MH, M, VL, VL
PT	4	3 × 2 (1 × 5)	ML, L, VL, Meet

Note: SE = Strength-Endurance, SP = Strength-Power, PT = Peak/Taper, VL = very light (65–70%), L = light (70–75%), ML = medium light (75–80%), M = medium (80–85%), MH = medium heavy (85–90%), H = heavy (90–95%), VH = very heavy (95–100%). Intensities are based off a set-rep best system [47]. Sets and reps in parentheses represent a single drop set at approximately 60% of the working sets.

**Table 3 sports-07-00129-t003:** Example Exercise Selection.

Day	Strength-Endurance	Strength-Power	Peak/Taper
Monday/Thursday	AM	AM	AM
	Back Squat	Back Squat	Back Squat*
		PM	PM
	PM	Push Press	Jerk
	Push Press	Jerk Lockout	Dead Stop Parallel Squat**
	Press from split	BTN Press	BTN Press
	DB Press	DB Press	DB Press*
Wednesday	AM	AM	AM
	Snatch Tech	Snatch Tech	Snatch Tech
	CGSS	CGSS	CGSS
	CG Pull–Floor	CG Pull–Floor	CG Pull–PP
	PM	PM	PM
	Snatch Tech	Snatch Tech	Snatch Tech
	CGSS	CGSS	SGSS
	CG Pull–PP	CG Pull–Knee	SG Pull–Floor
	CG SLDL	CG SLDL	CG SLDL*
	DB Row	CG Bent Over Row	DB Row*
Saturday	Snatch Tech	Snatch Tech	Snatch Tech
	SGSS	SGSS	SGSS
	Snatch	Snatch	Snatch
	C & J	C & J	C & J
	SG SLDL	SG SLDL	SG SLDL
	DB Row	SG Bent Over Row	DB Row

Note: DB = dumbbell, CG = clean grip, CGSS = clean grip shoulder shrug, SLDL = stiff legged deadlift, SG = snatch grip, SGSS = snatch grip shoulder shrug, BTN = behind the neck, C & J = clean and jerk. * Dropped during last week of taper. ** Only used during overreach (week 1).

**Table 4 sports-07-00129-t004:** Dependent Variables at Each Timepoint.

Variable	T1	T2	T3	T4	T5
PF (N)	4956 ± 1418	4942 ± 1499	4884 ± 1412	4948 ± 1378	4902 ± 1224
RFD50 (N·S^−1^)	2452 ± 1329	2392 ± 1820	2910 ± 1416	2503 ± 1290	3111 ± 1478
RFD100 (N·S^−1^)	5183 ± 3253	4808 ± 3455	6240 ± 3494	5379 ± 2325	6436 ± 3108
RFD150 (N·S^−1^)	7699 ± 4332	7112 ± 4170	8565 ± 4524	7852 ± 2999	8687 ± 4397
RFD200 (N·S^−1^)	8397 ± 3970	7850 ± 3853	9116 ± 3936	8465 ± 2955	8542 ± 3965
RFD250 (N·S^−1^)	7830 ± 3243	7450 ± 3226	8290 ± 2991	7917 ± 2261	7420 ± 2945
BM (kg)	71.9 ± 14.5	73.6 ± 15.5 †	73.2 ± 14.5 †‡	72.7 ± 14.3	72.7 ± 14.4
CSA (cm^2^)	39.2 ± 10.0	42.3 ± 10.1 †	41.0 ± 9.6 †‡	40.2 ± 9.9	40.1 ± 10.3
MT (cm)	2.82 ± 0.43	2.98 ± 0.43	2.88 ± 0.42	2.89 ± 0.42	2.88 ± 0.43
PA (°)	21.2 ± 5.45	21.5 ± 3.64	21.01 ± 5.16	19.9 ± 3.93	19.3 ± 4.89
FL (cm)	8.1 ± 1.9	8.2 ± 1.0	8.4 ± 2.1	8.7 ± 1.8	9.0 ± 1.3

Note: PF = Peak Force; CSA = Cross Sectional Area; MT = Muscle Thickness; PA = Pennation Angle; FL = Fascicle Length. † Significantly different from the previous timepoint (*p* ≤ 0.05). ‡ Significantly different from T1 (*p* ≤ 0.05).

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
