# Peer review of "Phase-Specific Changes in Rate of Force Development and Muscle Morphology Throughout a Block Periodized Training Cycle in Weightlifters"

_sports, 2019, doi:10.3390/sports7060129_

Reviewer 1 Report

The work is very interesting, however, there are several things to keep in mind. First of all, the number of subjects in the sample is very small, and it is also a very heterogeneous sample (men and women). On the other hand, the training program does not clarify if they have been phases of exactly 3 or 4 weeks. As far as statistical analysis is concerned, it is recommended to make a prior test of data normality, in order to know the most interesting statistical tool to use.
Therefore, it is recommended to review the article for possible publication:

You have 9 keywords, you should reduce to 4-5 maximum.

Line 19: the acronym "RFD" appears without exposing its meaning beforehand.

In the method, lines 97 and 98, phases 1 and 2 appear literally "consisted between 3 and 4 weeks". They must express exactly how long each training cycle has been temporary. 

The sample is very reduced to clarify results concludes
It is interesting to add decimals to the mean data in table 1.
In line 135: Explain better what the series and repetitions between parenthesis of table 2 mean.

You treat the results as parametric data, without having made a test on the normality of the data. It is recommended to make tests of normality, before treating the obtained data.
It is considered necessary to establish results by sex, as there may be different data in the tests conducted between men and women. And you take average data from the entire sample.

Author Response

Response to reviewer 1 comments:

Point 1: The work is very interesting, however, there are several things to keep in mind. First of all, the number of subjects in the sample is very small

Response 1: Thank you for your comments. We agree that this work and its results make for an interesting addition to the literature. The sample size of nine is small, but in the context of highly trained strength-power athletes, it is rare to find this subject number in previous research. Even a single subject design has been published with a subject of a similar level as this group (Bazyler et al. 2018).

Point 2: You have 9 keywords, you should reduce to 4-5 maximum.

Response 2: Keywords were reduced to 5. The instructions for authors on sports website states, “three to ten pertinent keywords need to be added after the abstract.”

Point 3: Line 19: the acronym "RFD" appears without exposing its meaning beforehand.

Response 3: Acronym is now addressed beforehand on line 19.

Point 4: In the method, lines 97 and 98, phases 1 and 2 appear literally "consisted between 3 and 4 weeks". They must express exactly how long each training cycle has been temporary.

Response 4: Lines 97-100 have been updated to express how long each phase lasted and match with figure 1 and table 2. Below are the updated sentences.

“The initial training phase (T1-T2) consisted of three weeks of high volumes and low to moderate relative intensities, termed a Strength-Endurance Phase (SE). The second phase of training (T2-T3) consisted of four weeks of moderate volumes at higher intensities, termed a Strength-Power Phase (SP). The final block of training (T4-T5) occurred at the very end of each macrocycle where the athletes underwent a single week of a sharp increase in volume (Overreach), followed by a three-week taper of low volume and moderate intensities, termed a Peak/Taper Phase (PT).”

Point 5: The sample is very reduced to clarify results concludes it is interesting to add decimals to the mean data in table 1.

Response 5: Data has been placed into decimal form to better clarify subject characteristics.

Point 6: In line 135: Explain better what the series and repetitions between parenthesis of table 2 mean.

Response 6: The following updated sentence is included to better explain this “Sets and reps in parentheses represent a single drop set at approximately 60% of the working sets.”

Point 7: You treat the results as parametric data, without having made a test on the normality of the data. It is recommended to make tests of normality, before treating the obtained data.

It is considered necessary to establish results by sex, as there may be different data in the tests conducted between men and women. And you take average data from the entire sample.

Response 7: A Shapiro Wilks test was used to determine normality. The original manuscript did not mention this because the data was normally distributed. Line 193-194 now states, “A Shapiro-Wilk normality test was used to verify if the data were normally distributed.”

We agree that in many cases, it is important to separate males and females due to certain physiological differences. However, in this study table 1 shows that the females in this group have higher peak force values than many previously published all male subject groups, and they had a higher overall training age than the males. Both their peak force and RFD values are comparable to male subjects of their size and therefore don’t warrant separation due to differences in strength level. Also, figure 5 effectively displays that the changes in muscle morphology followed similar trends in the individual subjects. The differences in force values and muscle size throughout this group were more due to size than sex. Therefore, we would not consider this group heterogeneous, and the results do not produce any differences due to sex making it unnecessary to report the results separately. Reporting of changes in these variables with trained males and females together has been done in previously published studies (Bazyler et al. 2017; Zaras et al. 2016).

Reviewer 2 Report

Suarez et al. analyzed the phase-specific changes in the rate of force development (RFD) and muscle morphology during a block periodized training cycle. This manuscript was well-written and well-organized.

Major points

1) Basically, all figures should be presented in the Results section. The authors should add the explanation for Fig. 4 and Fig. 5 in the Result section.

2) In line 195, the authors should describe the detail of statics. Which post hoc tests were used for analysis?

3) In Table.1, the authors should describe to the first decimal place.

4) In line 123, the authors should describe the approved number of this study.

Minor points

5) In line 19, the explanation for RFD was required.

Author Response

Response to reviewer 2 comments:

Reviewer 2:

1) Basically, all figures should be presented in the Results section. The authors should add the explanation for Fig. 4 and Fig. 5 in the Result section.

Response 1: The figures were originally placed in the results section and the current placement of the figures was made by the editor. Personally, we prefer the changes that the editor made but if it is not considered correct, placement of them back in their original position is also okay.

2) In line 195, the authors should describe the detail of statics. Which post hoc tests were used for analysis?

Response 2: Details of post hoc analysis were updated. Now states “Statistical effects were followed up with post hoc pairwise comparisons between time points. Effect sizes (Cohen’s d) and 95% confidence intervals were calculated to better provide population parameter estimates of mean change and to infer practically meaningful changes.” 

3) In Table.1, the authors should describe to the first decimal place.

Response 3: Data has been placed into decimal form to better clarify subject characteristics.

4) In line 123, the authors should describe the approved number of this study.

Response 4: Approved IRB # is now included on line 124.

5) In line 19, the explanation for RFD was required.

Response 5: Acronym is now addressed beforehand on line 19.

Round  2

Reviewer 1 Report

Ok, thanks for your modifications.

Author Response

Ok, thanks for your modifications.

Response: Thank you for your comments and suggestions. 

Reviewer 2 Report

2) In line 195, the authors should describe the detail of statics. Which post hoc tests were used for analysis?

Response 2: Details of post hoc analysis were updated. Now states “Statistical effects were followed up with post hoc pairwise comparisons between time points. Effect sizes (Cohen’s d) and 95% confidence intervals were calculated to better provide population parameter estimates of mean change and to infer practically meaningful changes.” 

The authors misunderstood my question. Because the authors performed post hoc analysis, they might perform Dunnet, Bonferonni or any other kinds of statistical analysis. The authors should describe which statistical method was used for post hoc analysis.

Author Response

Point: The authors misunderstood my question. Because the authors performed post hoc analysis, they might perform Dunnet, Bonferonni or any other kinds of statistical analysis. The authors should describe which statistical method was used for post hoc analysis

Response: Thank you for your feedback and sorry for the misunderstanding. The updated sentence now states, "Statistical effects were followed up with post hoc pairwise comparisons with Bonferroni adjustment.”